# Oral Health Status, Knowledge, and Behaviours of People with Diabetes in Sydney, Australia

**DOI:** 10.3390/ijerph18073464

**Published:** 2021-03-26

**Authors:** Prakash Poudel, Rhonda Griffiths, Amit Arora, Vincent W. Wong, Jeff R. Flack, George Barker, Ajesh George

**Affiliations:** 1Centre for Oral Health Outcomes and Research Translation (COHORT), Liverpool, NSW 2170, Australia; A.George@westernsydney.edu.au; 2School of Nursing & Midwifery, Western Sydney University, Campbelltown, NSW 2560, Australia; r.griffiths@westernsydney.edu.au; 3South Western Sydney Local Health District, Liverpool, NSW 2170, Australia; Vincent.Wong1@health.nsw.gov.au (V.W.W.); Jeff.Flack@health.nsw.gov.au (J.R.F.); 4Ingham Institute for Applied Medical Research, Liverpool, NSW 2170, Australia; 5School of Health Sciences, Western Sydney University, Campbelltown, NSW 2751, Australia; A.Arora@westernsydney.edu.au; 6Translational Health Research Institute, Western Sydney University, Penrith, NSW 2751, Australia; 7Discipline of Child and Adolescent Health, Sydney Medical School, Faculty of Medicine and Health, The University of Sydney, Westmead, NSW 2145, Australia; 8Oral Health Services, Sydney Local Health District and Sydney Dental Hospital, NSW Health, Surry Hills, NSW 2010, Australia; 9Faculty of Medicine, University of New South Wales, Kensington, NSW 2052, Australia; 10Diabetes Centre Bankstown-Lidcombe Hospital, Bankstown, NSW 2200, Australia; 11School of Medicine, Western Sydney University, Campbelltown, NSW 2560, Australia; 12Diabetes Education Service, Hornsby-Ku-ring-gai Hospital, Northern Sydney Local Health District, Hornsby, NSW 2077, Australia; George.Barker@health.nsw.gov.au; 13School of Dentistry, Faculty of Medicine and Health, University of Sydney, Camperdown, NSW 2050, Australia

**Keywords:** diabetes mellitus, oral health, oral health status, knowledge and behaviours, survey, diabetes care providers, public health, inclusive health

## Abstract

This study assessed self-reported oral health status, knowledge, and behaviours of people living with diabetes along with barriers and facilitators in accessing dental care. A cross sectional survey of 260 patients from four public diabetes clinics in Sydney, Australia was undertaken using a 35-item questionnaire. Data were analysed using SPSS software with descriptive and logistic regression analyses. More than half (53.1%) of respondents reported having dental problems which negatively impacted their related quality of life. Less than half (45%) had adequate oral health knowledge. Only 10.8% reported receiving any oral health information in diabetes care settings, which had higher odds of demonstrating adequate oral health knowledge (AOR, 2.60; 95% CI, 1.06–6.34). Similarly, 62.7% reported seeing a dentist in the last 12 months. Having private health insurance (AOR, 3.70; 95% CI, 1.85–7.40) had higher odds of seeing a dentist in the past 12 months. Dental costs were a major contributor to avoiding or delaying dental visit. Patients living with diabetes have unmet oral health needs particularly around the awareness of its importance and access to affordable dental services. Diabetes care providers can play a crucial role in this area by promoting oral health to their patients.

## 1. Introduction

In the past few decades, there has been a steady rise in the prevalence of Diabetes Mellitus (DM). In 2014, approximately 422 million people had DM worldwide [1]. In Australia, more than 1.3 million people were registered to the National Diabetes Services Scheme (NDSS) with DM in 2019, with an increase of over 100,000 new registrants over the previous 12 months [2]. Hyperglycaemia can cause complications related to most organ systems especially the eyes, kidneys, nerves, heart, and blood vessels [1]. Although not commonly discussed in diabetes care, even slightly elevated blood glucose levels adversely affect oral health, manifesting in several oral diseases and conditions [3], most commonly periodontal (gum) disease [4].

Periodontal disease affects a majority of the population worldwide, with the mildest form (gingivitis) affecting 50–90% of adults while the chronic stage (periodontists) affects 10–15% of the general population [3]. The increased risk for periodontal disease is reported to be two to three-fold higher for people with diabetes [3]. Furthermore, evidence show that diabetes and periodontal disease affect each other in a chronic and vicious cycle [3].

Poorly controlled diabetes affects periodontal outcomes and periodontitis also adversely affects blood glucose levels and worsens diabetes complications. The biological mechanism that links diabetes and periodontitis involves a complex interaction and that includes aspects of inflammation, immune functioning, neutrophil activity, and cytokine biology [5]. The evidence supports that uncontrolled diabetes causes to elevate levels of several pro-inflammatory mediators and cytokines in saliva and gingival crevicular fluid (GCF), oxidative stress in periodontal tissues and formation of Advanced Glycation Endproducts (AGE) [5]. Furthermore, the interaction of AGE– Receptor for Advanced Glycation Endproducts (RAGE) exaggerates inflammatory response (inflammatory dysfunction, cellular stress and other changes to important periodontal cells) and leads to periodontal tissue destruction [3]. Although evidence supports for a negative impact of periodontitis on diabetes control and outcomes, there is lack of mechanistic studies to explain its biological plausibility. However, potential factors include the mediators derived from periodontal disease (Interleukin (IL)-6 tumour necrosis factor (TNF)-α, and C-reactive protein (CRP) as well as oxygen radical) which impair insulin signalling and resistance [5].

There is also evidence that treatment of periodontal disease has beneficial effects on glycaemic control, with a reduction of glycated haemoglobin (HbA1c), although this evidence is often considered of low quality [6] due to the heterogeneity of the studies and small sample size [7,8]. Recognising the bidirectional link between diabetes and periodontal disease, as well as potential benefits of periodontal treatment, current guidelines [9,10] recommend that patients with diabetes optimise oral hygiene behaviours and seek regular oral health check-ups to prevent periodontal disease and maintain good oral health status [11,12]. Similarly, research indicates that improving oral health knowledge is essential for improving self-oral care practices [13].

Despite the bidirectional association between diabetes and periodontal disease and current recommendations, research from several countries report that people living with diabetes often have low levels of oral health knowledge, awareness, and compliance with good oral health behaviours [14]. However, in the Australian context, information about oral health knowledge, perceptions and practices of people living with diabetes is unknown. Therefore, gaining such information could help to inform preventative oral health measures for people living with diabetes in Australia. Hence, the aim of this study was to assess the oral health status, knowledge, and behaviours of people living with diabetes in Sydney, Australia. The study was guided by the following research questions:What are the perceived oral health status and self-reported dental problems among people living with diabetes?What are the oral health knowledge and behaviours of people living with diabetes?What are the factors associated with adequacy of oral health knowledge and dental visits among people living with diabetes?What are the perceived barriers and facilitators to accessing dental care?

## 2. Materials and Methods

### 2.1. Research Design

A cross sectional survey was conducted among people living with diabetes.

### 2.2. Sample and Setting

A convenience sample of 260 patients attending large public diabetes care centres in Sydney, Australia, were recruited between March 2019 and January 2020. These clinics have multidisciplinary diabetes care teams including endocrinologists, diabetes educators, dietitians and podiatrists who provide specialised care of patients with endocrine disorders or diabetes with complex medical issues generally referred by their general practitioners (GPs) [15]. The clinics were located at Liverpool, Fairfield and Bankstown-Lidcombe hospitals in South Western Sydney and Hornsby Ku-ring-gai hospital in Northern Sydney. These study sites were chosen in order to enable the recruitment of participants with diverse socio-economic background. Sydney South West local government area (LGA) includes disadvantages areas while the most advantaged areas are located in Northern Sydney LGA [16]. Furthermore, South Western Sydney has one of the highest rates of diabetes across Metropolitan Sydney [17].

The uptake of dental services by people with diabetes was used for the sample size estimation. However, due to the lack of information regarding the utilisation of dental services among people with diabetes in Australia, the uptake of dental services among people with diabetes globally was used to inform the sample size. Our systematic review study showed that around 54% of patients with diabetes had seen a dentist in the preceding year [14]. Therefore, assuming a conservative 50% dental attendance rate among people with diabetes, a total of 171 participants were required to estimate a 95% confidence interval of the proportion of people with diabetes who had seen a dentist in the previous year within 7.5% of the true population (margin of error). Allowing for 20% missing data, a sample size of 214 was required for this study.

### 2.3. Inclusion/Exclusion Criteria

People aged 18 years and over with a diagnosis of type 1 or type 2 diabetes were included in the study. Interpreters were utilised where possible for participants with limited English, and family members who were fluent in English were also asked to provide assistance to complete the questionnaire. However, those with inadequate English language to complete the questionnaire unaided and who did not have assistance of an interpreter or a family member at the time of data collection were excluded from the recruitment.

### 2.4. Data Collection

Flyers containing information about the study were distributed across waiting rooms of the study sites. A trained and experienced researcher (PP) provided an information sheet to potential participants, explained the purpose of the research and answered any queries. Participation was voluntary and those who met the inclusion criteria and agreed to participate in the study were provided with a consent form and self-administered questionnaire to complete while waiting for their medical appointment. Oral health information and dental products (toothbrush and toothpaste) were provided to all patients invited to take part in the study as a way of thanking them regardless of study participation. Written consent was obtained from all participants. It took between 10 and 15 min for participants to complete the questionnaire.

### 2.5. The Questionnaire Development and Pilot Testing

The study questionnaire was developed based on a review of existing literature [14,18] and previous research with GPs and diabetes educators [19,20]. The questionnaire consisted of 35 questions including the contextual and individual characteristics (predisposing, enabling and need), and health behaviours (personal health practices, process of medical care such as oral health information received from care providers, and use of hospital services such as dental visits) [21]. Item generation for the questionnaire was guided by Andersen’s behaviour model of health services use [21] to assess the factors influencing access to dental services among people with diabetes. These questions were grouped into seven domains that sought information on the subject’s perceived oral health status and oral health related quality of life (OHRQoL), knowledge about oral health, attitudes toward diabetes and oral health, oral health care practices and barriers to seeking dental care, information about diabetes care practices and oral health, family and social support, and demographic, socio-economic and health characteristics of the participants. The survey is found in Appendix A.

Face validity of the questionnaire was assessed by an expert panel consisting of clinicians, academics and educators in the field of dentistry, diabetes, and nursing (*n* = 6). Their comments on the survey items were sought through qualitative feedback and based on this, minor revision of items was undertaken. The questionnaire was then piloted with nine patients with diabetes to assess readability and relevance. Agreement was captured using yes (1) and no (0) on each item and calculated as percentage. Feedback was also sought to the questions where there was a disagreement. Questions that received less than 100% of agreement (*n* = 17) were revised accordingly.

### 2.6. Measures

The measures listed below formed part of the survey questionnaire and data analysis. Standardised questions that were validated to assess oral health status, knowledge and behaviours were used where available.

A single item question widely used in the previous studies [14,22] to assess overall oral health status (excellent, very good, good, fair and poor).A single item question to describe about dental problems (yes, no), with the list of most common problems found in people with diabetes [14].A 14-item validated oral health impact profile (OHIP-14) [23] questionnaire to assess OHRQoL.A 10-item oral health knowledge questionnaire, which included some validated items [13], to describe oral health knowledge levels.Oral health behaviours questions which were sourced from previous studies [14] to describe oral hygiene behaviours, dental visits, reasons for dental visits and source of oral health information.Demographic, socio-economic and health specific questions included age, sex, post code, marital status, education level, employment status, type and duration of diabetes, health insurance status, household income, and country of birth- to describe participant characteristics and enable comparisons.

### 2.7. Data Analysis

Data were analysed using Statistical Package for the Social Sciences (SPSS) Version 25 software [v.25, IBM, New York, NY, USA]. Descriptive statistics (mean and standard deviation for continuous variables and frequency counts and percentage for categorical variables) were used to present demographic, socio-economic and health specific characteristics, self-reported oral health status, OHRQoL, oral health knowledge and behaviours. For OHRQoL, OHIP-14 severity score was calculated by summing encoded responses (0 = never, 1 = hardly ever, 2 = occasionally, 3 = fairly often, 4 = very often) across 14 items, producing a range of values from 0 (best subjective oral health) to 56, with higher scores indicating worse OHRQoL. A threshold of “occasionally”, “fairly often” or “very often” as a response was used to determine an impact in OHRQoL [23]. For oral health knowledge, the median of the number of correct responses for the 10 knowledge items was calculated and adequate oral health knowledge was defined as achieving above the median score [13].

Binary logistic regression was used to examine factors associated with adequacy of oral health knowledge (model 1) and dental visits in the last 12 months (model 2). These explanatory models used covariates based on the evidence from previously published studies [14] and was informed by the Anderson model [21]. The variables included in the models involved individual predisposing and enabling characteristics (sex, country of birth, language spoken at home, educational attainment, private health insurance, oral health knowledge), contextual enabling factors (socio-economic status using the residential postcode as per the Index of Relative Socio-economic Advantage and Disadvantage (IRSAD) [16], and health behaviours (dental visit, self-reported oral health problems). Results were summarised as adjusted odds ratios (AORs) with 95% confidence intervals (CIs). Nagelkerke R2 was used to determine the strength of association of variables in the model, and the Hosmer-Lemeshow test to determine the goodness of fit of the model.

Some of the variables were recoded for analysis, which, for the most part, involved collapsing response categories (Appendix A). The recoding was done to break into dichotomous categories (country of birth, language spoken at home), merge the categories with less responses and/or to assist comparison with results from other studies (education, annual income and oral health status) and calculate the total score (oral health knowledge). The categories of ‘don’t know’ was treated as missing and removed from analysis.

## 3. Results

### 3.1. Demographic Charecteristics

A total of 281 patients were approached to participate in this survey and 260 completed the survey (92.5% response rate). The demographic, socio-economic and health specific characteristics of people living with diabetes are summarised in Table 1. The mean (±SD) age of the participants was 61.7 (±13.8) years and over half (53.5%) of the participants were male. Majority (68.1%) were born overseas and just above half (57.7%) spoke English language at home. More than three quarters (86.1%) of the participants had secondary or tertiary level education. Further, over half (55.0%) had a combined household income of less than AUD 40,000. While reporting socio-economic status using the residential post codes of the participants as per IRSAD, just under one third (30.8%) of the participants were living in the most disadvantaged areas.

In relation to type of diabetes, most of the participants (86.9%) reported having type 2 diabetes. The median numbers of years since the diagnose of diabetes was 13 (range < 1 to 60 years). More than two thirds (68.1%) of participants did not have private health insurance, with over a half (58.0%) eligible for public dental services either via holding a health care card or being a member of the defence force.

### 3.2. Oral Health Status

Table 2 summarises the findings of self-reported oral health status, knowledge and behaviours. More than half of the participants (54.3%) reported their oral health status as good to excellent, however, they also reported having one or more oral health problems (53.1%). The most common oral health problems included dry mouth (23.8%), gaps between teeth (23.1%), pain in teeth and or gums (21.9%) and loose teeth (21.9). OHIP-14 score showed that a majority of the participants (71.2%) reported their oral condition had impacted on at least one of the seven domains (score of 2 “occasionally” or higher), with physical pain being the most impaired (64.2%) and social disability the least (38.8%). The mean (SD) OHIP 14 score for participants was 11.38 (±12.0).

### 3.3. Oral Health Knowledge

The mean (± SD) knowledge score of the participants was 5.2 (±2.5) out of a total score of 10. Less than half of the participants (45%) had adequate oral health knowledge with a score of more than 5 (the median score) (Table 2). Moreover, poor knowledge (26.5–46.2%) was observed around the items related to the bidirectional link of diabetes and oral health and benefits of dental treatment on blood glucose management (item 1,2 & 5) (Appendix A). However, participants did have sound knowledge around good oral hygiene practices such as flossing (item 10) (80.8%). Only 10.8% of the patients reported receiving oral health information from their diabetes care providers.

### 3.4. Oral Health Behaviours

More than half (62.7%) of the participants reported seeing a dentist in the last 12 months and the majority of visits were for dental problems or treatments (59.4%), followed by check-up or teeth cleaning (Table 2). Less than a third of patients reported receiving advice in relation to benefits of checking blood glucose (23.5%) and quitting tobacco (31.5%) from their dentist in their last visit. In terms of oral hygiene behaviours, two thirds of respondents (67.7%) reported brushing their teeth/dentures twice or more a day, with an overwhelming majority (92.3%) using fluoride toothpaste. However, most respondents were not using a floss or an interdental brush daily.

### 3.5. Predictors of Having Adequate Oral Health Knowledge

Two factors were found to be associated with significant and independent predictors of having adequate oral health knowledge. Those who received oral health information from diabetes care providers had 2.6 times higher odds of demonstrating adequate oral health knowledge (AOR, 2.60; 95% CI, 1.06–6.34) than those who did not receive information. Similarly, participants with technical or further education (TAFE) (AOR, 3.17; 95% CI, 1.14–8.77) and University level education (AOR, 2.96; 95%CI, 1.12–7.82) had 3 times higher odds of demonstrating adequate oral health knowledge than those with up to primary level education (Table 3). These factors explained 14.5% of the total model variance (Nagelkerke R^2^ = 0.145). The Hosmer-Lemeshow goodness-of-fit test had a χ^2^ of 9.691 (df = 8, *p* = 0.287), indicating a good model fit.

### 3.6. Predictors of Having Dental Visit in the Last 12 Months

As shown in Table 4, results of the binary logistic regression analysis showed four variables as significant and independent predictors of having seen a dentist in the previous 12 months. Participants with private health insurance were almost four times higher odds of seeing a dentist in the last 12 months (AOR, 3.70; 95% CI, 1.85–7.40) than those who did not have insurance. Similarly, participants with university level education had almost three times higher odds of seeing a dentist (AOR, 2.98; 95% CI, 1.11–8.00) than those with up to primary level education. Compared to the participants who were born in Australia, those born overseas had two times higher odds of seeing a dentist in the last 12 months (AOR, 2.16; 95% CI, 1.13–4.12). Similarly, females had almost two times higher odds of seeing a dentist (AOR, 1.82; 95% CI, 1.04–3.20) than their counterpart. These factors explained 16.2% of the total model variance (Nagelkerke R^2^ = 0.162). The Hosmer-Lemeshow goodness-of-fit test had a χ^2^ of 13.683 (df = 8, *p* = 0.090), indicating a good model fit.

### 3.7. Barriers and Facilitators in Seeking Dental Care

Among respondents who did not visit the dentist in the last 12 months, the most common reasons included cost of dental care (60.1%) and lack of perceived need for dental care (41.2%) (Table 5). Furthermore, nearly half (47.3%) of the respondents reported that they did not have financial support to see a dentist or have dental treatment if necessary. However, in terms of the support, more than three quarters (77.7%) reported that they had family or friends to talk about their oral health problems and get their support when they have oral health problems (81.2%), as well as attending dental appointment if needed (79.2%). Similarly, a majority of them (87.3%) also reported that they had easy access to transport if they needed to attend a dental appointment (Appendix A).

## 4. Discussion

To the best of our knowledge, this is the first study to assess the self-reported oral health status, knowledge and behaviours of people living with diabetes in Australia. The survey recruited diverse participants from across the socio-economic spectrum in Sydney. The sample recruited was fairly representative of the population data from the Australian National Diabetes Audit (ANDA) in terms of their mean (SD) age 61.7 (±13.8) vs. 56.0 (±17.7), median duration of their diabetes 13.0 vs. 15.3 [15], and proportion of respondents with type 2 diabetes. Further, more than two-thirds of the respondents were born overseas (68.1%), which was a different from the ANDA data, but this is not surprising considering that most (88%) were recruited from South Western Sydney which is the most culturally diverse community in NSW with more than half (52.7%) of the population born overseas [24].

A large proportion of respondents rated their oral health status as poor to fair (45.8%), and this was higher than the Australian general population (24%) [25], which is understandable since diabetes predisposes people to various oral health problems [3]. This finding was also higher than those reported in Canada (21%) which has a similar healthcare system [26]. A possible reason for this variation could be the presence or absence of diabetes related chronic or acute complications in the respondents, as complications have been found to be associated with greater frequency of reporting poor oral health status [26]. In our study, half of the respondents reported to have other chronic diseases in additional to diabetes and the majority had a lower socio-economic status, so these factors would have played an additional role in the higher prevalence of poor oral health status. More than half of the patients reported having one or more oral health problems with dry mouth being the most prevalent issue and this was similar to the findings of a study from Netherlands [27]. However, the mean (SD) OHIP-14 severity score (11.38 ± 12.0) found in our study was significantly higher compared to that reported among participants in the Dutch study (2.5 ± 5.2) [27]. Higher OHIP severity score denotes a greater impact in OHRQoL. The impact of oral conditions affecting in one or more subdomains was also higher (71.2%) compared to the Dutch study (19%) [27]. The higher OHIP scores observed in our study is possibly because fewer people had seen a dentist in the last 12 months. Nevertheless, it is clear that patients with diabetes are in need of oral health care support as it is negatively impacting their quality of life.

In keeping with the previous research [14], more than half of the participants were found to have inadequate oral health knowledge related to diabetes most notably, around the bidirectional link of diabetes and oral health. This is consistent with our systematic review which concluded that individuals with diabetes have lower oral health knowledge than those without diabetes [14]. Consistent with earlier studies, our results showed that receiving oral health information [13,28] from care providers and a higher level of educational attainment [29] were significant predictors of having adequate oral health knowledge irrespective of socio-economic status. Our previous review also highlighted that patients who were better informed or had good knowledge of the link between diabetes and oral health were more likely to adopt optimal oral health behaviours [14]. Despite these findings, very few participants in our study (10.8%) reported receiving oral health information from DCPs, which is a common phenomenon reported in the literature [14]. A probable reason for this could be that promoting oral health among patients in the diabetes care settings is a new and often challenging task due to the time constraints as well as limited knowledge and confidence to promote oral health as they are not adequately skilled or trained in aspects of oral health [19,20]. Therefore, specific training or clinically focused program to build DCPs’ capacity in oral health care and define their role is required, as studies report patients receiving oral health information from DCPs are more likely to maintain optimal oral health behaviours including regular dental visits [14].

Considering the time constraints of DCPs, a short oral health risk assessment tool needs to be developed along with an education and training program to capacity build DCPs in this area [19,20]. As part of the model of care, GPs or endocrinologists should be encouraged to conduct an annual periodontal review of patients with diabetes and provide referral to those identified or at risk to periodontal disease [9,10]. The role of certified diabetes educators (CDE) in this area should also be explored as part of an intraprofessional model of care [19], as they see more patients with diabetes than other diabetes care providers in Australia [15] and have been found to be very supportive in promoting oral health among patients [19]. While considering the bidirectional link of diabetes and periodontal disease, an interprofessional collaboration between oral health care professionals (OHCPs) and DCPs is essential [3].

Whilst exploring oral health behaviours, two-thirds of patients reported brushing two times a day and most (92%) reported using a fluoridated toothpaste. However, the practice of interdental cleaning appeared to be least important for the patients, as nearly half (47%) reported that they did not use a floss or interdental brush in the last 7 days. The lower compliance on this aspect of oral self-care among the patients is consistently reported in previous studies [14]. One of the possible reasons could be that the importance for interdental cleaning has not been promoted effectively to the public. A study conducted among the general population in the USA reported that the lower compliance in interdental cleaning is due to poor literacy and lack of awareness [30]. Furthermore, patients also reported seeing fewer advertisements for floss in comparison to tooth brushes and toothpaste [30]. Studies have found the absence of interdental cleaning to be negatively associated with blood glucose control and oral health problems [31,32]. Further, there is some evidence that interdental cleaning in addition to regular brushing, may reduce gingivitis or plaque, or both. Thus, it is clear that, at the very least, there needs to be greater communication about the importance of regular flossing with this at risk population [33]. 

Visits to a dentist in the past 12 months was also reported to be lower (62.7%) compared to UK (85.2%) [34], Sweden (85.1%) [35], Netherland (76%) [27], and USA (72.7% [36] and 65.8% [37]). Few overseas studies, which compared the dental attendance, also showed that the rate is lower in patients with diabetes than those without diabetes [35,36,37,38]. This is of great concern, as we know diabetes increases the risk of periodontal disease, which in turn compromises glycaemic control and worsens diabetes related complications [3,4]. The lower rate of dental visits among this population could be due to the priority given to other health issues that patients with diabetes are often dealing with such as, depression, mental health treatment, and diabetes distress [39]. The lack of private health insurance, especially when a significant proportion of our study population belonged to lower socio-economic groups, could also contribute to lower frequency of dental visits [35,36,37,38]. Previous studies reported a strong association between private health insurance and dental visits [40,41], and our study also demonstrated that those with private health insurance were almost four times more likely to see a dentist in the last 12 months. Over a third of our respondents who did not see a dentist in the last year reported such avoidance or delay was due to the cost of dental care, which is also widely reported elsewhere [14]. Although more than half with welfare cards were eligible for government-subsidised dental care through public dental services, long waiting times is commonly reported as a significant barrier to accessing public dental care in Australia [42,43]. These challenges to dental care have also been a significant barrier for DCPs to promote oral health and encourage patients to seek dental treatment [19,20]. 

Delaying dental care for more than one year is a concern for this at risk population, as a study conducted in USA involving a nationally representative sample (*n* = 70,363) found that this was associated with increased odds of worse self-rated health and greater physical unhealthy days compared to those who visited a dentist in the past year [44]. These findings underscore the need of a prioritised referral pathway for patients with chronic diseases which may facilitate timely access and utilisation of dental care services. Early treatment not only reverse dental disease and rehabilitate the teeth and gums but also prevent unnecessary hospitalisation for dental related conditions, which was reported to be about 72,000 in the year 2017–2018 in Australia [25].

Like previous studies [38,40], another strong predictor of dental visits in the last 12 months was higher educational level. This could relate to the fact that people with higher education levels have greater knowledge of oral health disease and treatment, and hence would also have higher compliance of optimal oral health behaviours. On the other hand, our study showed that those born overseas and female patients were more likely to have seen the dentist in the last 12 months, which is also consistent with other studies conducted across various clinical care settings [43,45]. A possible explanation for higher dental attendance among the overseas born population could be due to the ‘healthy immigrant effect’, widely reported in the literature [46]. This suggests that the overseas born population may come with better health including oral health status than the host country. However, their health tends to deteriorate over the years of living in the host country [47]. As such, a Canadian study indicated an increase of self-reported dental problems among immigrants within 2 years of arrival [46]. Another study which analysed the national survey data in the USA found that females had a greater odds of visiting a dentist in the past 12 months [45]. The increased access of females to health services in USA is discussed in relation to the concept of feminisation (gender) of health access [48,49], such as females appeared to be accessing health services including oral health care more often, partly because of prenatal care, commonly acting the family caregiver duties [48] and also, they are more concerned about their appearance as well as about the fear of future pain [49]. However, further research needs to be conducted to explore the utilisation of dental services among females in Australia and determine the predictive factors. 

When interpreting the findings, it is important to consider a number of limitations. Firstly, this study consists of a convenience sample that were recruited from tertiary diabetes care services, which provides specialist assessment and treatment. Therefore, the participants were more likely to have poorer glycaemic control with increased comorbidities and complications [15] and this may not be an accurate representation of the diabetes population in Australia. Nevertheless, the oral health status, behaviours and knowledge of the patients in the present study are more or less similar to those reported in the general population. Further, our study also consisted of a relatively large sample size and a good response rate. Another potential limitation is that we asked respondents about their visit to a dentist so it is probable that they might have seen other oral health care professionals such as oral health therapist. Similarly, we also did not ask the patients about the type of dental practice (public/private) they visited in the last year. A further limitation is that the data are self-reported and hence subject to recall bias and also social desirability bias, that is, if a respondent believes that one should see a dentist annually, they might report a dental visit even if one does not occur [45]. Finally, the use of only close-ended questions (i.e., yes/no/don’t know) in the oral health knowledge test may allow participants to guess the correct answer. Despite these limitations this study has provided a valuable insight into this under researched area in Australia.

## 5. Conclusions

People living with diabetes in Australia have self-reported poor oral health status and it impacts their oral health related quality of life. Yet, many have poor oral health knowledge and are not engaging in optimum oral health behaviours, particularly around regular interdental cleaning and dental visits. Further, people with diabetes are not receiving oral health information from DCPs to improve their knowledge and are experiencing significant barriers in terms of costs to access dental care. This warrants the need of oral health preventive strategies in routine diabetes care. Such strategies could include oral health education, risk assessment and referral activities as part of the diabetes care model. To support this mode of care, health promotional resources and a clear referral pathway should be explored to facilitate patient education and uptake of dental services, respectively. Training programs to capacity build DCPs to provide education to patients about oral health problems and optimal oral health behaviours, along with risks assessment and dental referrals are required. A key part of this model will need to be accessible and affordable dental services to facilitate referrals and uptake of dental care among patients. An intraprofessional collaboration among the diabetes care providers is required to carry out oral health promotion activities in the diabetes care settings.

## Figures and Tables

**Table 1 ijerph-18-03464-t001:** Demographic, Socio-economic and Health specific characteristics *N* = 260.

Variables			Frequency (%)
Demographic	Age	(mean, (SD) range)	61.7 (13.8) 19–96
	Sex *	Male	139 (53.5)
	Country of Birth	Australia	83 (31.9)
	Language spoken at home	English	150 (57.7)
	Marital Status *	Married	162 (62.3)
		Widowed	34 (13.1)
		Divorced	32 (12.3)
		Single	31 (11.9)
Socio-economic	Highest educational attainment	Up to Primary schooling	36 (13.9)
		Secondary schooling	108 (41.5)
		Tertiary	116 (44.6)
	Employment status	Not working	190 (73.1)
	Annual Household income	<AUD 40,000	143 (55.0)
		AUD 40,000 to < 60,000	27 (10.4)
		AUD 60,000 to < 80,000	21 (8.1)
		AUD 80,000 to <100,000	9 (3.5)
		100+	25(9.6)
		Don’t know	35 (13.5)
	Index for Relative Socio-economic Advantage and Disadvantage (IRSAD) 1 = most disadvantaged 10 = most advantaged	Deciles 1 and 2	80 (30.8)
		Deciles 3 and 4	65(25.0)
		Deciles 5 and 6	35 (13.5)
		Deciles 7 and 8	36 (13.8)
		Deciles 9 and 10	44 (16.9)
Health specific	Type of Diabetes	Type 2	226 (86.9)
		Type 1	29 (11.2)
		Don’t know	5 (2.0)
	Duration of diabetes	(median, (IQR) range)	13 (14) 0-60
	Recent blood glucose test(self-reported)	Too high	76 (29.2)
		About right	177 (68.1)
		Too low	2 (.8)
		Don’t know	5 (1.9)
	Private Health Insurance	Yes	83 (31.9)
	Eligible for public dental service	Yes	151 (58.1)
	Other co-morbidities	Yes	130 (50.0)
	Smoking	Yes	37 (14.2)
	Drinking	Yes	74 (28.5)

* Missing data (range 1–2).

**Table 2 ijerph-18-03464-t002:** Self-reported oral health status, knowledge and behaviours.

Variables		Frequency (%)
Oral Health Status	Good to excellent	141 (54.3)
	Poor to fair	119(45.8)
OHIP-14	OHIP-14 severity score mean (SD)	11.38 (12.0)
	Impact on ≥ 1 subdomains	185 (71.2)
	Most affected subdomain- physical pain	167 (64.2)
	Least affected subdomain social disability	101 (38.8)
Oral Health Problems **	Yes	138 (53.1)
	Main oral health problems/concerns (*n* = 138) **	
	Dry mouth	62 (23.8)
	Gaps between teeth	60 (23.1)
	Pain in teeth and/or gums	57 (21.9)
	Loose teeth	57 (21.9)
Oral Health Knowledge	Score mean (SD)	5.2 (±2.5)
	Adequate (>5)	45
	Inadequate (<5)	55
Dental visit in the last 12 months	Yes	163 (62.7)
	Main reason behind visiting the dentist (*n* = 163) *	
	Dental problems & treatment	97 (59.4)
	Check-up/exam/Cleaning	91 (55.8)
Advice from dentist in the last visit	Checking your blood sugar	
	Yes	61 (23.5)
	Giving up cigarettes or other types of tobacco	
	Yes	82 (31.5)
Brushing frequency/day	Twice a day or more	176 (67.7)
	Once a day	64 (24.6)
	A few times a week	15 (5.8)
	Never	5 (1.9)
Oral hygiene products used **	Fluoride toothpaste	240 (92.3)
	Mouthwash	86 (33.1)
	Sugar free chewing gum	31 (11.9)
	None	6 (2.3)
Use dental floss/interdental brush in the last 7 days *	0 day	122 (46.9)
	1–3 days	75 (28.8)
	4–6 days	15 (5.8)
	7 days	47 (18.1)
Received oral health information from DCPs	Yes	28 (10.8)

* Missing data (range 1–2), ** multiple responses.

**Table 3 ijerph-18-03464-t003:** Predictors of having adequate oral health knowledge.

Variables		Univariate Analysis (Unadjusted)				Multivariate Analysis (Adjusted)			Overall
	N	Odds Ratio	95% CI	*p* Value	Overall *p* Value	Odds Ratio	95% CI	*p* Value	Overall *p* Value
OH from DCPs *									
Not received	232	1.00	(Reference)			1.00	(Reference)		
Received	27	2.42	1.07–5.47	0.034		2.60	1.06–6.34	0.036	
Education *					0.02				0.026
Up to primary	36	1.00	(Reference)				(reference)		
Secondary/High school	107	1.84	0.79–4.29	0.161		1.44	0.60–3.50	0.425	
TAFE	49	4.35	1.69–11.20	0.002		3.17	1.14–8.77	0.027	
University	67	3.93	1.61–9.63	0.003		2.96	1.12–7.82	0.029	

OH = Oral Health DCPs = Diabetes Care Providers, TAFE = Technical and Further Education, CI = Confidence Interval. Variables entered on this model: received oral health information, IRSAD index, sex, place of birth, education, dental Visit, and language spoken at home, * missing data (*n* = 1).

**Table 4 ijerph-18-03464-t004:** Predictors of dental visit in the last 12 months.

Variables	N	Univariate Analysis (Unadjusted)				Multivariate Analysis (Adjusted)			Overall
		Odds Ratio	95% CI	*p* Value	Overall *p* Value	Odds Ratio	95% CI	*p* Value	Overall *p* Value
Private Health Insurance *									
Without PH insurance	175	1.00	(Reference)			1.00	(Reference)		
With PH insurance	83	3.24	1.76–5.96	0.000		3.70	1.85–7.40	0.000	
Education *					0.007				0.034
Up to Primary	36	1.00	(Reference)				(reference)		
Secondary/High school	106	1.12	0.52–2.40	0.770		1.10	0.48–2.48	0.837	
TAFE	49	0.90	0.38–2.14	0.820		0.92	0.34–2.46	0.860	
University	67	3.32	1.36–8.12	0.008		2.98	1.11–8.00	0.030	
Sex *									
Male	139		(Reference)				(Reference)		
Female	119	1.44	0.87–2.40	0.158		1.82	1.04–3.20	0.037	
Place of birth *									
Australia	83		(Reference)				(Reference)		
Overseas	175	1.46	0.85–2.48	0.167		2.16	1.13–4.12	0.020	

PH = Private Health, TAFE = Technical and Further Education CI = Confidence Interval. Variables entered on this model: sex, place of birth, education, oral health information received from DCPs, oral health knowledge, IRSAD index, private health insurance and oral health problems. * missing data (range 1–2).

**Table 5 ijerph-18-03464-t005:** Main reasons for not visiting a dentist in the last 12 months ** (*n* = 97).

Reasons **	Frequency (%)
Cost related	62 (60.1)
Did not have any dental problems	40 (41.2)
Nothing serious/expected dental problems to go away	14 (14.4)
Fear	13 (13.4)
Too busy/unable to take off time from work	10 (10.2)
Inconvenience	3 (3.0)
Other	12 (12.3)

** multiple responses.

## Data Availability

Data are available from the corresponding author upon reasonable request.

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
