# Peer review of "Oral Health Status, Knowledge, and Behaviours of People with Diabetes in Sydney, Australia"

_ijerph, 2021, doi:10.3390/ijerph18073464_

Round 1

Reviewer 1 Report

The data present in this research are relevant and offer a good point of view but the article should be rewritten to be more clear and scientifically sound. 

In particular

The template used must be eliminated from the main text; correct the text

The abbreviation can be used only after being written in long form with the abbreviation in square brackets.

The paragraph (line 63-66) should be rewritten to underline the poor scientific relevance of the findings. 

The introduction should be deeply revised  focusing on the relationship between the diabetes and parodontal disease.

In the material and method  the authors should state the inclusion and exclusions criteria. 

Authors should divide the statistical analysis performed from the data analysis parameters. 

The parameters evaluated should be stated in the matherial section.

The discussion should be simplified focusing on the main findings of the research and comparing it to the one retrieved from the literature. 

The reference 21 is wrong and the details should be positioned in the text in square brackets.

The reference section should be revised formatting it following the journal guidelines. (ex. 40)

Author Response

Dear Reviewer 1:

Thank you very much for reviewing this paper. Below is an outline of how we have addressed your comments. The amended sections have been highlighted (with dark background) in the revised manuscript.

Response to the reviewer 1

Comment 1

The data present in this research are relevant and offer a good point of view but the article should be rewritten to be more clear and scientifically sound. 

Response: We have revised the manuscript substantially as per the reviewers’ feedback.

Comment 2:

The template used must be eliminated from the main text; correct the text

Response: We have checked with the journal editorial team about this and were advised to inform you that this was due to network issues and has already been corrected.

Comment 3:

The abbreviation can be used only after being written in long form with the abbreviation in square brackets.

Response: We kindly advise you that we have proof-read the paper and used the acronyms correctly throughout the revised manuscript.

Comment 4:

The paragraph (line 63-66) should be rewritten to underline the poor scientific relevance of the findings. 

Response: We acknowledge the reviewer’s comment about the effectiveness of periodontal treatment on glycaemic control is of lower quality. We did highlight this in the previous statement please see page 3, lines 77-79, which reads as below:

“There is also evidence that treatment of periodontal disease has beneficial effects on glycaemic control, with a reduction of glycated haemoglobin (HbA1c) although this evidence is often considered of low quality due to the heterogeneity of the studies and small sample size”.

To further highlight the poor scientific relevance of the findings, we have made a minor edit to the highlighted sentence. Please see p.3, lines 80-81. The section reads as below:

Recognising the bidirectional link between diabetes and periodontal disease, as well as potential benefits of periodontal treatment…

Comment 5:

The introduction should be deeply revised focusing on the relationship between the diabetes and periodontal disease.

Response: We acknowledge the reviewer’s suggestion. As advised, we have added a paragraph describing the link between diabetes and periodontal disease. Please see p.2 lines 61-74.

Comment 6:

In the material and method, the authors should state the inclusion and exclusions criteria. 

Response: As suggested, we have refined this section and added a subheading- inclusion/exclusion criteria. Please see section 2.3 in p.4, lines 127-134

Comment 7 & 8:

Authors should divide the statistical analysis performed from the data analysis parameters. The parameters evaluated should be stated in the matherial section.

Response: As suggested, we have added a new section 2.6 Measures and discussed about the parameters evaluated. Please see section 2.6 in pp.4-5 lines 169-187.

We have also improved the clarity of the analysis section. Please see the data analysis section p.5, lines 189-193.

Comment 9:

The discussion should be simplified focusing on the main findings of the research and comparing it to the one retrieved from the literature. 

Response: As advised, we have shortened several sections of the discussion to focus on the main findings and comparison with literature. Please see highlighted sections in:

p.9 lines 332-335

p10 lines 336-351 and 365-381

p11, lines 414-416 and 434

p12 lines 473-475

Comment 10:

The reference 21 is wrong and the details should be positioned in the text in square brackets. The reference section should be revised formatting it following the journal guidelines. (ex. 40)

Response: As advised, we removed the reference 21 and have placed the details in the text (please see p.5 line 190). We have also formatted the references as per the Journal guidelines.

Reviewer 2 Report

Dear authors,

Thank you for submitting your manuscript!

The aim of this study was to assess the oral health status, knowledge, behaviors of diabetic patients in Sydney, Australia. The overall objective is good, and this is a well-structured paper. However, the manuscript needs some adjustments.

General Comments:

  1. Please make sure that you write the reference in the right place and follow the appropriate sentence structure.
  2. Please check the English language/spelling through the entire paper.

Introduction:

  1. Please add a reference to this sentence “The increased risk for periodontal disease is reported 57 to be two to three-fold higher for people with diabetes.”

Materials and Methods:

  1. I suggest that sections 2.1, 2.2 and 2.5 to be combined.
  2. Please state why some of the variables were recoded and how.

Results:

  1. In Table 3, do you think it is a fair comparison to compare the group who received oral health information or with the one who did not since the big difference in the participants number in each group?!
  2. Was the number of participants who received oral health information 27 or 28? Please check the discrepancy in reporting.
  3. Please reference the tables inside the text.
  4. I suggest moving section 3.5 before section 3.4.
  5. There is a discrepancy between table 1 and table 4 in reporting participants with private health insurance.
  6. I recommend adding a table for section 3.7 even if the data was reported in the supplementary materials. There is no limit for the tables number.

Discussion and Conclusion:

  1. Please avoid repetitions in ideas and sentences.
  2. Please combine the conclusions. There are multiple paragraphs for conclusion in both discussion and conclusion sections.

Thank you!

Reviewer 3 Report

  1. What was the justification behind selecting a convenience sample instead of random sampling methodology? A convenience sampling methodology induces very high risk of bias owing to the few centers from which patients were recruited to the study.
  2. Why was no sample power estimation done prior to recruitment? It is simply mentioned that 281 patients received the surveys, out of which 260 responded.
  3. Please mention the units of continuous variables such as duration of diabetes and blood glucose levels (if they were not HbA1c%) in the table 1.

Author Response

Dear Reviewer 3

Thank you very much for reviewing this paper. Below is an outline of how we have addressed the your comments. The amended sections have been highlighted (with dark background) in the revised manuscript.

Response to the Reviewer 3

Comment 1

What was the justification behind selecting a convenience sample instead of random sampling methodology? A convenience sampling methodology induces very high risk of bias owing to the few centers from which patients were recruited to the study.

Response:

We kindly advise that this study was part of a doctoral program. The investigators had time constraints and limited resources and hence utilised convenience sampling. However, all attempts were made to recruit a representative sample by choosing large diabetes centres across affluent and disadvantaged locations of Sydney. Further, South Western Sydney was chosen because it has one of the highest rates of Diabetes in metropolitan Sydney and we have added this information (please see p.3, lines 115-116). Further, we had calculated/estimated the sample size for the survey and achieved the required sample size. Please see p.3, lines 117-126.

Nevertheless, we do acknowledge the limitations of convenience sampling and have highlighted this in the limitations section of the discussion. Please see page 12, lines 450-456.

Comment 2:

Please mention the units of continuous variables such as duration of diabetes and blood glucose levels (if they were not HbA1c%) in the table 1.

Response:

As suggested, we have added the unit for duration of diabetes (year). We kindly advise that the questionnaire didn’t collect the information about HbA1c- and unit, rather collected self-reported response with the options of too high, about right and too low. Please see table 1 in p.6.

Comment 3:

Why was no sample power estimation done prior to recruitment? It is simply mentioned that 281 patients received the surveys, out of which 260 responded.

Response:

We kindly advise that the sample size was calculated prior to recruitment and is described in the manuscript. Please see section 2.2 sample and setting in p. 3 lines 117-126.

Round 2

Reviewer 1 Report

The revision is well made by the authors, now the paper is readable and scientifically sound.  

Reviewer 3 Report

Thank you for addressing the review comments in the revises manuscript.